# Viability of Glycolysis for the Chemical Recycling of Highly Coloured and Multi-Layered Actual PET Wastes

**DOI:** 10.3390/polym15204196

**Published:** 2023-10-23

**Authors:** Asier Asueta, Sixto Arnaiz, Rafael Miguel-Fernández, Jon Leivar, Izotz Amundarain, Borja Aramburu, Jose Ignacio Gutiérrez-Ortiz, Rubén López-Fonseca

**Affiliations:** 1GAIKER Technology Centre, Basque Research and Technology Alliance (BRTA), Parque Científico y Tecnológico de Bizkaia, Edificio 202, 48170 Zamudio, Bizkaia, Spain; arnaiz@gaiker.es (S.A.); miguel@gaiker.es (R.M.-F.); leivar@gaiker.es (J.L.); amundarain@gaiker.es (I.A.); aramburu@gaiker.es (B.A.); 2Chemical Technologies for Environmental Sustainability Group, Chemical Engineering Department, Faculty of Science and Technology, University of the Basque Country (UPV/EHU), 48940 Leioa, Bizkaia, Spain; joseignacio.gutierrez@ehu.eus (J.I.G.-O.); ruben.lopez@ehu.eus (R.L.-F.)

**Keywords:** household packaging waste, plastic waste, highly coloured PET, multi-layered PET, on-line identification, NIR, solvolysis, catalytic glycolysis, BHET

## Abstract

The chemical recycling of poly(ethylene terephthalate) –PET– fractions, derived from actual household packaging waste streams, using solvolysis, was investigated. This recycling strategy was applied after a previous on-line automatic identification, by near-infrared spectroscopy –NIR–, and a subsequent selective sorting of the different PET materials that were present in the packaging wastes. Using this technology, it was possible to classify fractions exclusively including PET, virtually avoiding the presence of both other plastics and materials, such as paper, cardboard and wood, that are present in the packaging wastes, as they were efficiently recognised and differentiated. The simple PET fractions, including clear and monolayered materials, were adequate to be recycled by mechanical means meanwhile the complex PET fractions, containing highly coloured and multi-layered materials, were suitable candidates to be recycled by chemical routes. The depolymerisation capacity of the catalytic glycolysis, when applied to those complex PET wastes, was studied by evaluating the effect of the process parameters on the resulting formation and recovery of the monomer bis(2-hydroxyethyl) terephthalate –BHET– and the achieved quality of this reaction product. Comparable and reasonable results, in terms of monomer yield and its characteristics, were obtained independently of the type of complex PET waste that was chemically recycled.

## 1. Introduction

Nowadays, due to the global awareness about environmental issues in the society, the optimisation of technologies for the treatment and recycling of plastic wastes is crucial [1]. Demand for poly(ethylene terephthalate) –PET– worldwide has almost doubled during the last years, from 17 million metric tonnes in 2010 to near 29 million metric tonnes in 2021 [2], being soft-drinks (27.1%) and mineral water (34.6%) bottles the most demanded products [3]. In parallel to this increased consumption, the waste generation and its collection, sorting and recycling, via the implemented waste management systems that responds to the, by law, enforced extended producer responsibility, have exhibited similar growing trends. The global market of recycled PET was estimated to be USD 8.9 billion in 2021 and is projected to reach USD 11.7 billion by 2026. This growth is due to the rising demand to manufacture bottles, sheets, trays, fibres, straps and other products throughout the world. An increasing trend in terms of collection, reuse, recycling, recovery of PET wastes, and implementation of legislation requesting the use of recycled materials have been witnessed in the market for recycled PET –rPET–. These activities have consequently resulted in a growing demand for rPET in various applications, such as food and beverage bottles, textile fibres and strapping [4]. As for Europe, PETCORE, the European association of organisations involved in the PET value chain, announced that the amount of sorted PET bottles from collected waste streams for re-processing reached 2.0 million tonnes in 2018, an increase of about 5% related to previous year’s data, and this trend will continue [5,6].

In addition to the demand of the material itself, the need for the packaging sector of improved content preservation has been responsible for the development of barrier properties by combining materials in multi-layered products. On the other hand, the marketing strategies have favoured the use of highly coloured materials to create distinctive brand images and increase perceived quality of contained products. One emerging challenge associated with PET recycling is to provide a suitable treatment of those highly coloured and complex multi-layered materials in post-consumer waste fractions of PET. Both fractions are not convenient to be mixed with clear or slightly coloured PET, sorted to be mechanically recycled, and hence an alternative to downcycling or disposal is necessary. The complex PET fractions are less appropriate for mechanical recycling because derived products present serious limitations in either colour and transparency or intrinsic viscosity, limiting their value and usage. In accordance with this, by working with actual complex PET wastes, the present work aims to address the viability of chemical recycling as an alternative to the current management for the non-mechanically recoverable plastics, even though this option has not begun to be applied until recent dates only covered the treatment of 0.4% of the total plastic waste collected in Europe [7].

In this work near infrared –NIR– spectroscopy, an on-line identification technique able to recognise materials and used to sort waste streams, was applied to separate waste PET bottles and packages into two fractions: one containing slightly coloured and monomaterial PET waste, to be mechanically recycled, and another one including highly coloured and complex multi-layered PET to be managed by chemical recycling. NIR spectroscopy is a suitable technology for automatic identification of polymeric materials such as wood, plastics, foils, paper, etc., without any previous preparation by just recording and comparing their spectra with known references and has been selected because it combines real-time analysis with qualitative identification [8,9]. Currently, household packaging waste sorting facilities implement NIR spectroscopy for separating only mechanically recoverable PET. Hence, the separation of chemically recoverable PET, which would be a significant improvement and result in a lower amount of refused material, needs to be tuned in terms of both adjusted sorting parameters to deal with the specific characteristics of the waste and fulfilled requirements of a material to enter the chemical recycling process. The study of the PET glycolysis process was conducted after completing the conditioning of the reactor feed, the actual complex PET wastes, that comprises dirtiness removal and comminution steps. The reaction consists of the degradation of polymer and usually is conducted at temperatures in the range of 180–240 °C, atmospheric pressure, with an excess of ethylene glycol –EG– and the presence of a catalyst, typically zinc acetate, which leads to the formation of bis(2-hydroxyethyl) terephthalate –BHET– [10,11]. In principle, glycolysis conditions allow for the use of reduced amounts of reagents as well as lower operation temperatures and pressures when is compared with other chemical recycling methods such as methanolysis or thermal degradation [12,13,14,15]. Likewise, hydrolysis under alkaline or acid conditions involve serious technical and environmental issues associated with corrosion and management of the liquid effluents [16,17]. BHET, after an adequate purification, can be used either for PET synthesis or the production of other polyesters, polyurethanes, plasticisers, epoxy resins, and additives for textiles and biocompatible materials [18,19,20,21,22,23]. These numerous applications, in turn, provide an economical flexibility when using the raw recycled monomer for the most profitable option upon demand.

## 2. Experimental

### 2.1. Materials

In this work a reference clear PET, grade A according to internal standards for recycled PET defined by ECOEMBALAJES ESPAÑA, S.A. (ECOEMBES), Madrid, Spain, the Spanish management system for light packaging waste, and samples of highly coloured and multi-layered post-consumer complex PET wastes provided, also by ECOEMBES, were used. Data related to these PET materials are included in Table 1.

All post-consumer PET waste from bottles and packages were ground to 5 mm flakes and in order to avoid the impact of dirtiness on reaction tests, further cleaned with a diluted aqueous sodium hydroxide solution (1 wt%), rinsed with water and dried at 80 °C overnight in order to mimic the industrial treatment undergone by the reference clear PET. Clean PET flakes were again milled with a cryogenic mill to adjust the particle diameter to 250 µm in order to ensure a large available surface area and minimise mass and heat transfer limitations on the measured reaction rates. The solvent, EG (ethylene glycol 99%), was purchased from Aldrich and the catalyst, Zn(OAc)_2_·2 H_2_O (zinc acetate dihydrate, 99.5%), was provided by Probus. In order to simulate reaction conditions expected to be found in industrial chemical recycling plants, reagents were directly used without any further purification.

### 2.2. Automatic Identification Tests

A programmable commercial automatic identification and sorting system, the UniSort^®^ (RTT Systemtechnik GmbH, Zittau, Germany), was used for the post-consumer waste PET packages identification and sorting tests. The system includes an acceleration belt, an identification belt, a sensor head with 32 probes, two nozzle groups for classification by blowing air, a collecting hood, and an operation and control unit. Identification tests were carried out with representative samples of light packaging post-consumer wastes in order to evaluate both separation and sorting rates. The infrared sensor lines measure the complete spectral region, allowing the fast identification of polymeric materials. Even so, fast contact-free measurement under reflection, analysis without sample preparation, and the ability to measure fast moving samples are the main advantages of this system [24]. NIR spectra ranging from 4000 to 10,000 cm^−1^ (1000–2500 nm) are characterised by broad bands and low molar absorption coefficients. The absorption in this range results from harmonics of the fundamental frequencies in the middle infrared –MIR– region, activated because of their anharmonicity. Hence, NIR technology is especially suited for the detection of samples with –C-H, -N-H and -O-H bonds.

### 2.3. Glycolysis of PET

Glycolysis consists of a depolymerisation reaction of the PET polymer and more specifically a transesterification between PET ester groups and a diol, usually ethylene glycol –EG–, to obtain the monomer, in that case, the bis(2-hydroxyethyl terephthalate) –BHET– following the Equation (1) and as shows Figure 1:(PET)_n_ + (n − 1) EG ⇔ n (BHET)(1)

As aforementioned, the reaction is operated at, relatively, mild conditions, temperature below 200 °C and atmospheric pressure, with the presence of metal cations, as Zn^2+^ or Na^+^, dispersed in the media, that are added in the form of metallic salts and act as catalysts [25]. The catalytic glycolysis reaction is accepted to occur through the nucleophilic attack of the hydroxyl group of the diol on the carboxylic group present in the polyester structure. The attack is activated by the metal cation, via formation of a coordination complex between the carboxylic group and the metal that reduces the electronic density of the carboxylic group and facilitates the nucleophilic attack on the positively polarised carbon atom. The strength of the metal-oxygen bond of the carboxylic group determines the specific catalytic activity of a given cation. The shorter the bond, the stronger the interaction, which is consistent with the higher level of activity of the Zn^2+^ when compared with the Na^+^ since it can cause a greater charge polarisation that improves the glycolic attack.

Glycolysis experiments were carried out in a 500 cm^3^, a three-necked, flat-bottom glass reactor that was electrically heated and equipped with a thermometer and a reflux condenser operated at atmospheric pressure. A magnetic stirrer is incorporated to the tank to ensure proper mixing and turbulence. In all runs, 30.0 g of PET together with 1% catalyst, Zn(OAc)_2_, by weight basis were loaded into the reactor. The reaction vessel and the mixture of EG reactant and the catalyst were preheated to the selected temperature (165–196 °C) prior the addition of the PET particles in order to minimise the time required by the solid to reach the reaction temperature. The extent of the reaction was studied by determining the amount of BHET obtained in the product mixture after quenching the reactor vessel and its content. Firstly, a set of reactions with reference clear PET was carried out to analyse the effects of reaction time, temperature and EG/PET ratio and select the appropriate conditions for the depolymerisation of PET fractions including post-consumer highly coloured or multi-layered complex PET wastes. Table 2 summarises the experimental conditions of the runs carried out in this work. The reaction mixture was allowed to react for time intervals ranging from 0.33 up to 8 h. The following procedure was employed to recover and quantify the BHET yield. Hot distilled water was then added in excess to the reaction mixture with vigorous agitation since BHET is known to be quite soluble in boiling water. Hot water also dissolves the catalyst employed, and probably to some extent higher oligomers such as dimers and trimers. While still hot, the suspension was quickly filtered. The product was separated into solid and aqueous phases using a sintered glass filter (Whatman glass microfiber binder free, grade GF/C-1.2 µm) under vacuum. The glycolysed product was obtained as a residue after filtration. The filtrate contained EG, BHET, and small quantities of few water-soluble oligomers. This was heated until a clear mixture was obtained. Afterwards, it was filtrated with a sintered glass filter (Whatman glass microfiber binder free, grade GF/C-0.7 µm) under vacuum. This second filtrate was collected and stored in a refrigerator at 5 °C for 16 h to provoke the precipitation of white crystalline BHET flakes. After filtration, BHET was dried in an oven at 60 °C for 30 h, and weighed on an analytical balance to estimate the yield according to the following equation:(2)$$Y(\%)=\frac{W_{\;BHET,f}/M_{\;BHET}}{W_{\;PET,0}/M_{\;PET}}\times 100$$

In Equation (2) *W _PET,0_* and *W _BHET,f_* refer to the initial weight of PET and the weight of BHET at a specific reaction time, respectively. *M _BHET_* and *M _PET_* are the molecular mass of BHET (254.24 g mol^−1^) and the PET (192.17 g mol^−1^) repeating unit, respectively. This purified product was then subjected to various characterisation techniques.

### 2.4. Analysis of Glycolysis Products/Characterisation of BHET

Elemental analysis was carried out with an EuroVector Elemental Analyzer apparatus (EuroVector, the Elemental Analysis Company, Redavalle, Italy). Differential scanning calorimetry –DSC– was performed on a Mettler Toledo DSC822e instrument (Mettler-Toledo Ltd., Port Melbourne, Australia), with 3–5 mg samples, under a purge gas flow of 15 cm^3^ min^−1^ with nitrogen, and with a heating rate of 10 °C min^−1^ in the range 25–350 °C. A FTIR Nicolet Protegé 460 spectrometer (Thermo Fisher Scientific Inc., Waltham, MA, USA) was used in transmission mode with a resolution of 2 cm^−1^ to record spectra from depolymerisation products on KBr discs (100 mg with a dilution of 1/50, 13 mm in diameter). The instrument was equipped with a DTGS detector averaging 50 scans. Proton nuclear magnetic resonance –^1^H NMR– spectroscopy analysis was recorded with a Brucker AV500 spectrometer (Brucker Corp., Billerica, MA, USA) operating at 500 MHz. The spectra were obtained in d_6_-acetone solution.

## 3. Results and Discussion

### 3.1. Automatic Identification Tests

Representative samples of the plastic fraction coming from actual household packaging waste streams collected by ECOEMBES were studied by means of a NIR spectrometer. The NIR spectra of both plastic (PET, high-density polyethylene –HDPE–, polyvinyl chloride –PVC–, acrylonitrile butadiene styrene –ABS–, polypropylene –PP– and polystyrene –PS–) and non-plastic materials (wood, paper and cardboard), usually found due to their application for packaging, were recorded and examined. As shown in Figure 2 and Figure 3, PET spectrum presents a distinctive peak at 1657 nm, which made it possible to differentiate packaging materials, including this polymer, from the others.

On-line identification of the PET fraction and subsequent automatic sorting, with high yield and purity, was successful using the UniSort^®^ system. Recognition of highly coloured and multi-layered PET materials was not possible by NIR alone. Image analysis and colour sorting technology is proposed for the segregation of those fractions in a second separation step.

### 3.2. Glycolysis Reactions

Time, temperature and EG/PET molar ratio were chosen as the independent variables to be analysed for the glycolysis process, keeping constant the catalyst/PET ratio at 1% by weight. Run #1 was performed using reference PET for studying the evolution of BHET yield with time. Reaction progress at 0.33, 0.66, 1, 2, 3, 4, 5 and 8 h at 196 °C and with an EG/PET molar ratio of 7.6/1 is shown in Figure 4. Run #2 was used to compare the reaction rate in the absence of any catalyst, which was found to be rather slower. Although Run #1 revealed that the catalysed reaction reached equilibrium after 2 h, in order to shorten the duration of preliminary work, a reaction interval of 1 h was selected to determine the influence of the independent variables, such temperature and EG/PET molar ratio. Runs #1, #3 and #4 were performed at 165, 180 and 196 °C, respectively, and confirmed the need for operating at the maximum reachable temperature that, in practice, corresponds to the boiling point of EG at atmospheric pressure. The achieved BHET yields at a given time interval (1 h) with temperature are included in Figure 5. Finally, Runs #5 and #6 corresponded to runs with a lower EG/PET molar ratio (3.8/1 and 5.7/1). It was found that a large excess of EG (7.6/1) was required for attaining a reasonable BHET yield as plotted in Figure 6.

As the last step of this study, the glycolysis process of clear, coloured (green, amber, blue, white and silver), and multi-layered recycled PET were preliminarily compared in Figure 7 (Runs #7 to #12). In the literature, few reports can be found regarding the management of these types of wastes by glycolysis [26]. The experimental conditions used were 196 °C, EG/PET molar ratio 7.6/1 and 1 wt% Zn(OAc)_2_ after a reaction time interval of 2 h. It could be noticed that, within the experimental error, these seven types of recycled PET yield practically similar glycolysed products with conversion ranging between 59–68%. Therefore, the additives present in the recycled coloured PET apparently did not affect the extent of depolymerisation, although it tends to discolour the glycolysed products, unless an extra extraction step is taken to isolate it.

### 3.3. Analysis of Glycolysis Products

In order to check the purity of the recovered BHET after glycolysis, this product was analysed by means of a number of analytical techniques, and results were compared with those corresponding to a commercial BHET provided by the Aldrich Chemical Co. The results of the elemental analysis, expressed as the molar percentages of C, H and O, of the purified BHET product were 56.4% C, 5.4% H and 38.2% O, in fairly good agreement with 57.1% C, 5.9% H and 37.0% O found for the commercial sample. As for ^1^H NMR analysis, signals at δ 8.1, 4.4, 4.2 and 3.9 ppm were noticed. The signal at δ 8.1 ppm indicated the presence of the four aromatic protons of the benzene ring. Signals at δ 4.4 and 3.9 were characteristic of the methylene protons of COO-CH_2_ and CH_2_OH, respectively (Figure 8). On the other hand, the triplet at δ 4.1 ppm was attributed to the protons of the hydroxyl group [27]. In addition, the FTIR spectra (Figure 9) of the purified monomer clearly showed -OH band at 3450 cm^−1^ and 1135 cm^−1^, C=O stretching at 1715 cm^−1^, alkyl C-H at 2879 and 2954 cm^−1^ and aromatic C-H at 1411–1504 cm^−1^, present in BHET [28]. Further, the DSC scan also showed a reasonably sharp endothermic peak at 110 °C in agreement with the known melting point of BHET (Figure 10). From all these observations, it was concluded that the purified monomer was BHET, irrespective of the PET waste glycolysed.

On the other hand, DSC analysis of the glycolysed product (the residue obtained after filtration of the hot reaction mixture) was performed. As an example, the profile of the sample corresponding to Run #1 is included in Figure 11. Two distinct peaks were clearly noticed at about 160 and 220 °C. They were associated with the presence of the oligomers in the sample. Particularly, the peak located at 160 °C was attributed to the melting point of the dimer while the peak at 220 °C was reasonably assigned to the melting point of a mixture of oligomers having between 3–5 repeating units [29]. The absence of oligomers having a higher molecular weight was confirmed. This observation was in line with the possibility of controlling the selectivity of the process towards BHET by using a large excess of EG, this is to say, operating with a high EG/PET molar ratio [30]. The peak at 110 °C corresponds to traces of BHET, which were not properly removed during the filtration step at high temperature. It must be pointed out that the characteristic DSC peak corresponding to unreacted PET was not observed. This suggests that under equilibrium conditions only BHET and oligomers are present in the reaction mixture. According to Wang et al. [31] it is believed that PET with polymerisation degree of n was initially into PET with polymerisation degree of m (m < n). Afterwards, partially degraded PET was converted into oligomers, which were further depolymerised into the dimer and ultimately the BHET monomer. Under these conditions, the subsequent polymerisation of BHET into the dimer or higher oligomers also takes place.

## 4. Conclusions

On-line NIR identification tests of PET materials for their subsequent automatic sorting were carried out by means of a commercial system. Results revealed that NIR was a feasible technology for sorting a PET fraction from other non-polymer and polymer based materials present in packaging waste streams but failed on the recognition between highly coloured and multi-layered PET fractions. This task will require image analysis and/or colour sorting technologies for its segregation.

The glycolysis of reference PET was conducted under atmospheric pressure, with an excess of ethylene glycol as glycolytic agent and zinc acetate dihydrate as catalyst. The influence of glycolysis time, temperature and EG/PET molar ratio was investigated, showing that conversion depended on these factors significantly. The main products were BHET and a residual amount of oligomers. Once defined the most suitable process conditions, highly coloured and complex multi-layered post-consumer PET wastes were depolymerised, giving BHET yields up to about 70%. These values were similar to that obtained when treating reference PET, thereby suggesting that PET nature and colour had a minor influence on BHET yield. DSC and FTIR analysis verified that the BHET product derived from highly coloured and complex multi-layered post-consumer PET was equivalent to that obtained from reference PET. As a first step involved in a more ambitious research project, glycolysis was demonstrated to be a promising industrial strategy for the recycling of these non-mechanically recoverable actual PET wastes, which can probably be chemically recycled by glycolysis as a mixture of highly coloured and complex PET materials.

## Figures and Tables

**Figure 1 polymers-15-04196-f001:**
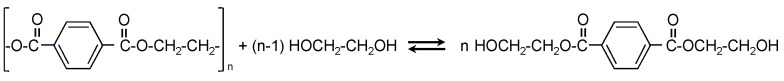
Depolymerization of PET to produce BHET using EG as glycolysis agent.

**Figure 2 polymers-15-04196-f002:**
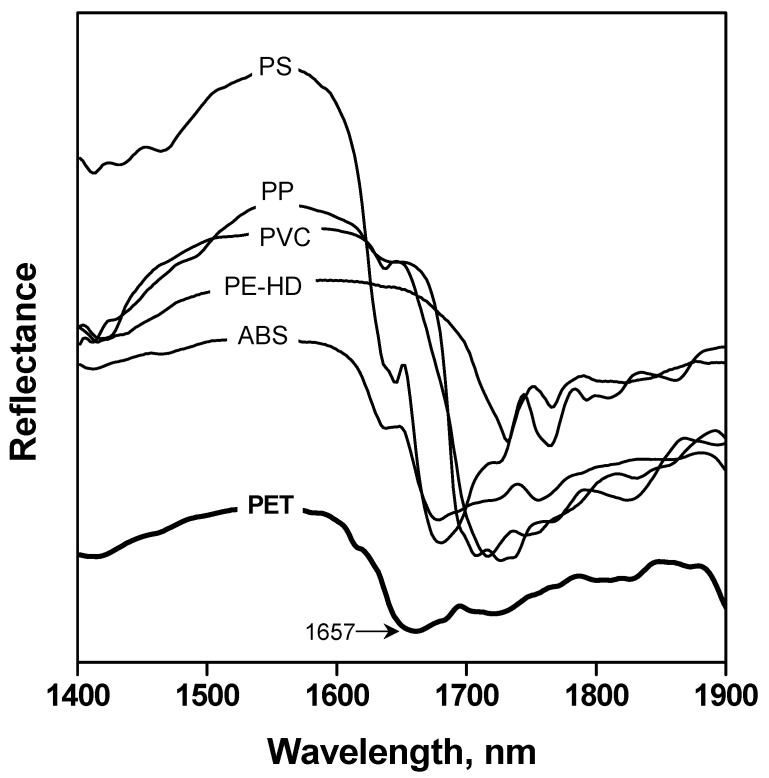
NIR spectra of PET and other polymers.

**Figure 3 polymers-15-04196-f003:**
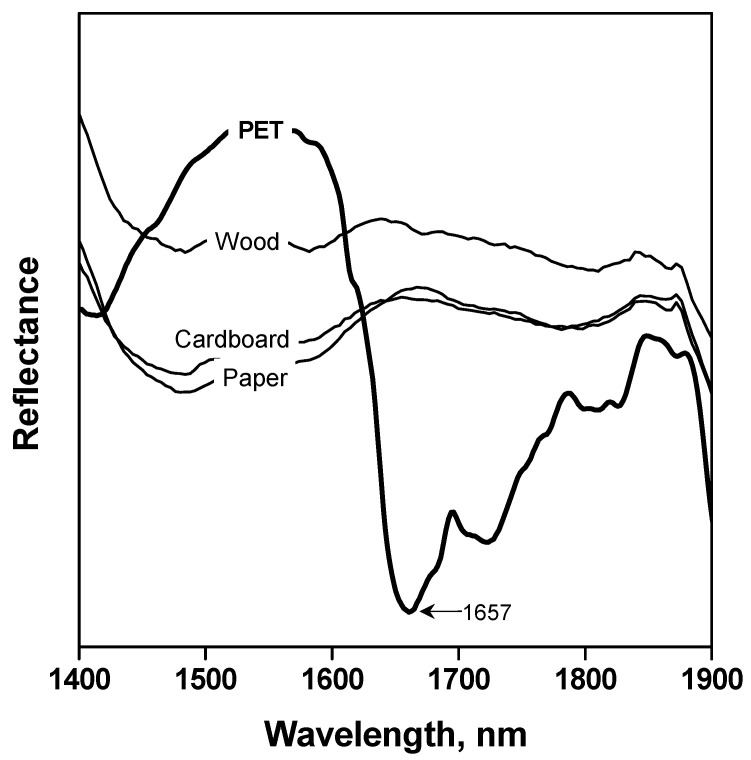
NIR spectra of PET and other no-polymers wastes.

**Figure 4 polymers-15-04196-f004:**
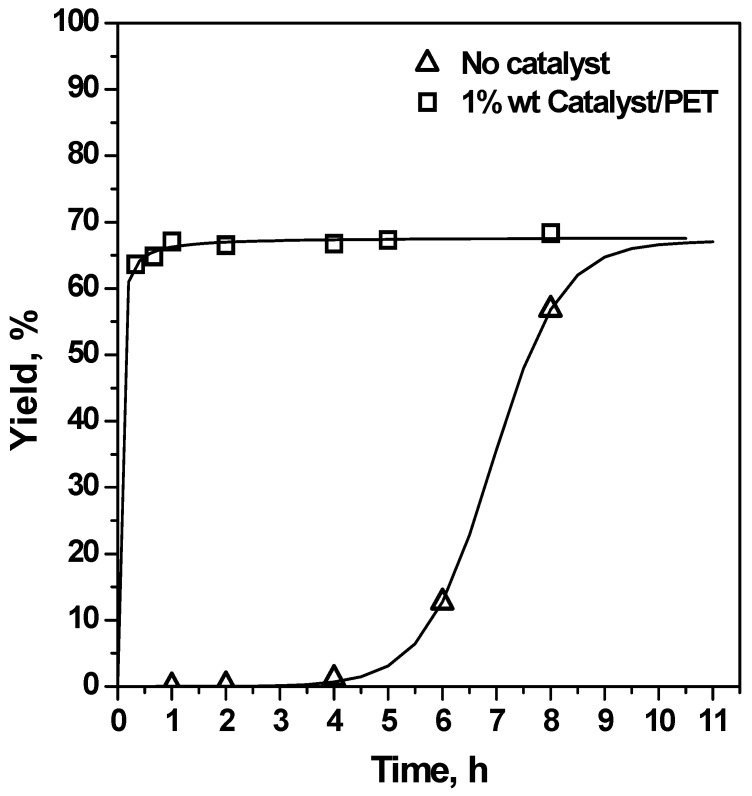
Effect of reaction time on BHET yield (reference PET, 196 °C, EG/PET = 7.6/1, Runs #1 and #2—squares correspond to experimental data from catalytic runs; triangles correspond to experimental data from non-catalytic glycolysis).

**Figure 5 polymers-15-04196-f005:**
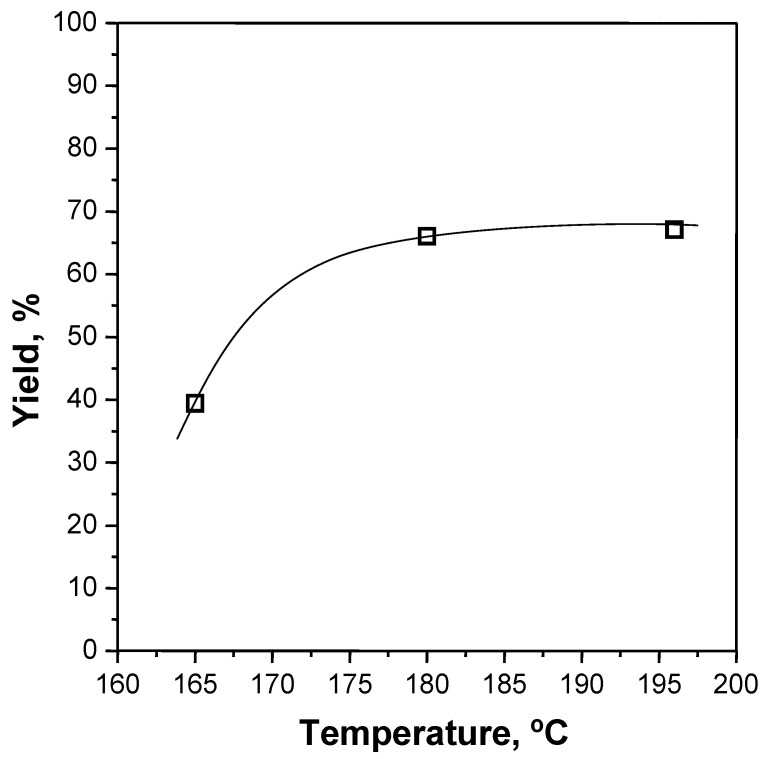
Effect of temperature on BHET yield (reference PET, EG/PET = 7.6/1, 1 h, 1 wt% Zn(OAc)_2_, Runs #1, #3 and #4).

**Figure 6 polymers-15-04196-f006:**
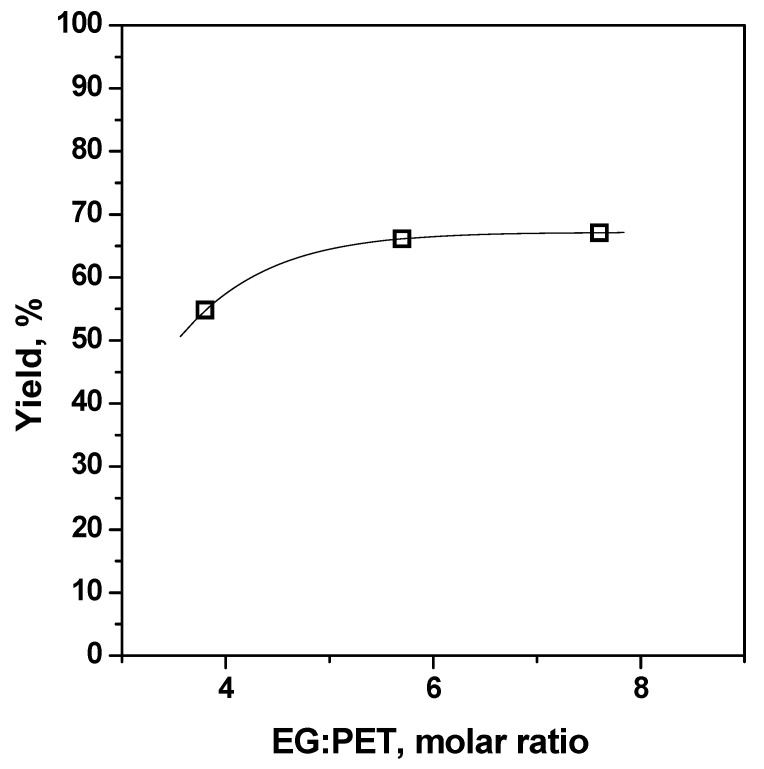
Effect of PET/EG mol ratio on BHET yield (reference PET, 196 °C, 1 h, 1 wt% Zn(OAc)_2_, Runs #1, #5 and #6).

**Figure 7 polymers-15-04196-f007:**
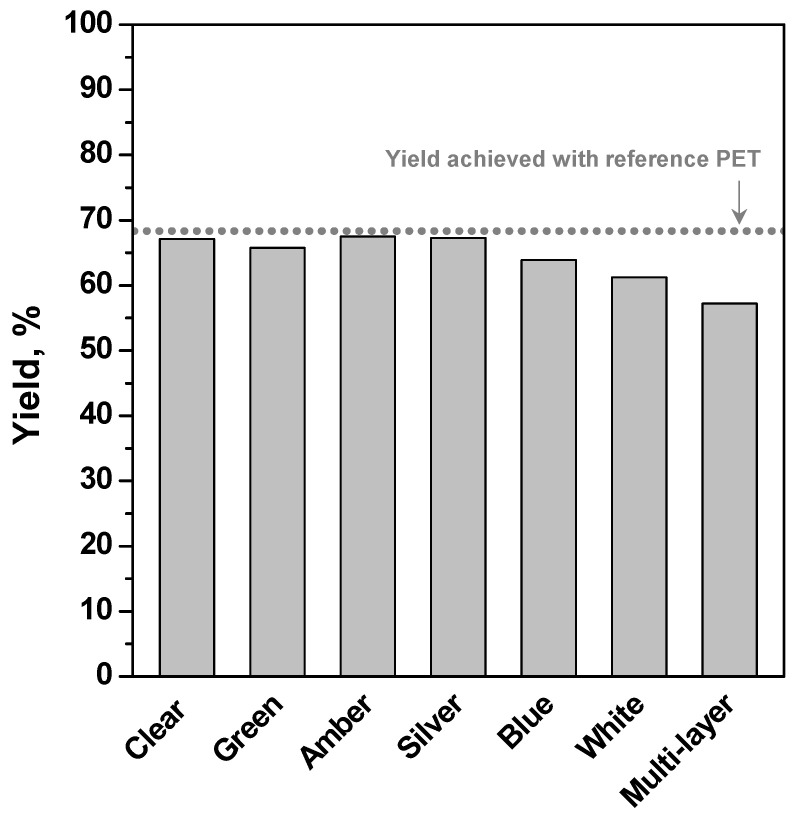
Effect of the nature of the PET waste on BHET yield (196 °C, EG/PET = 7.6/1, 2 h, 1 wt% Zn(OAc)_2_, Runs #7–12).

**Figure 8 polymers-15-04196-f008:**
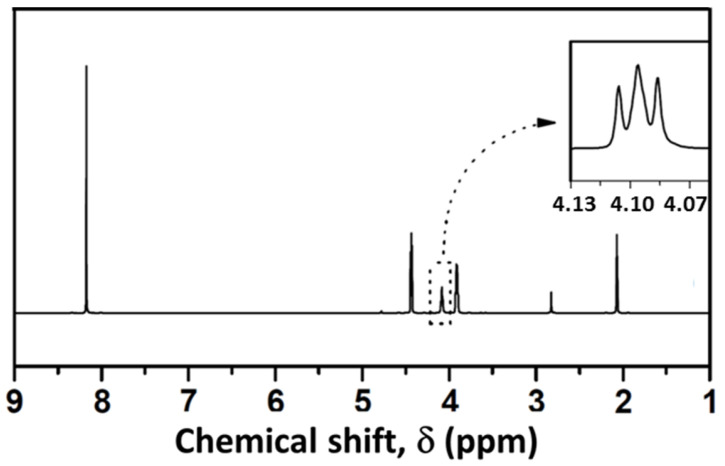
^1^H NMR spectra of BHET in d6-acetone.

**Figure 9 polymers-15-04196-f009:**
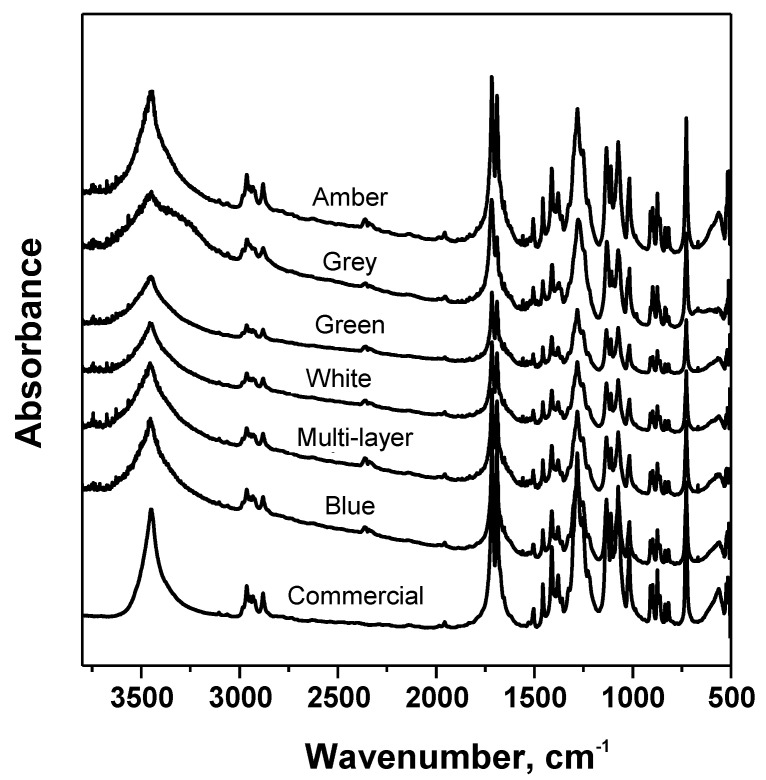
IR spectra of BHET from glycolysis of highly coloured and multi-layer PET wastes.

**Figure 10 polymers-15-04196-f010:**
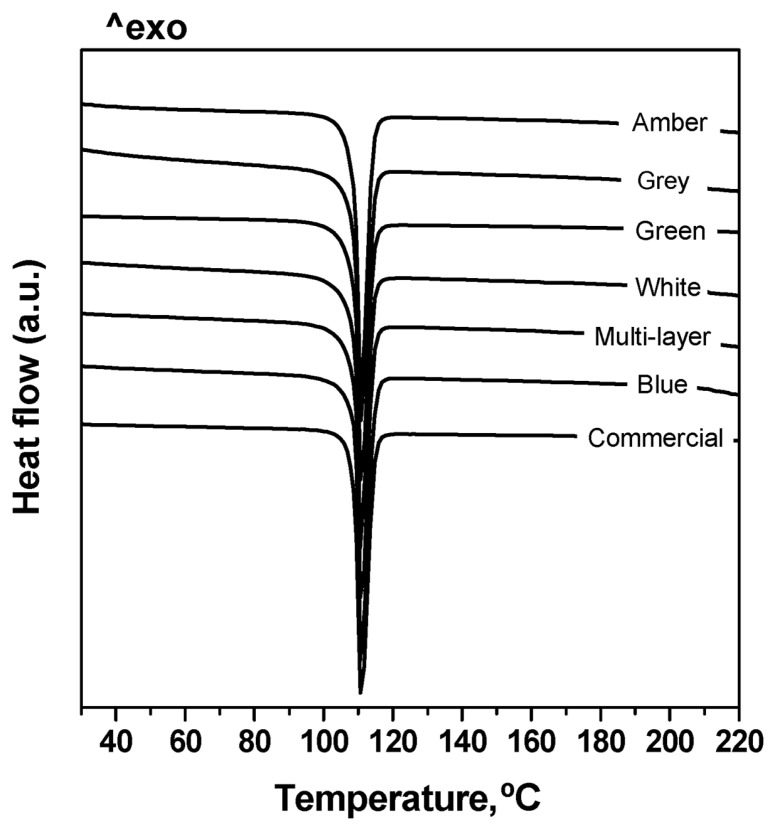
DSC scans of BHET from glycolysis of highly coloured and multi-layer PET wastes.

**Figure 11 polymers-15-04196-f011:**
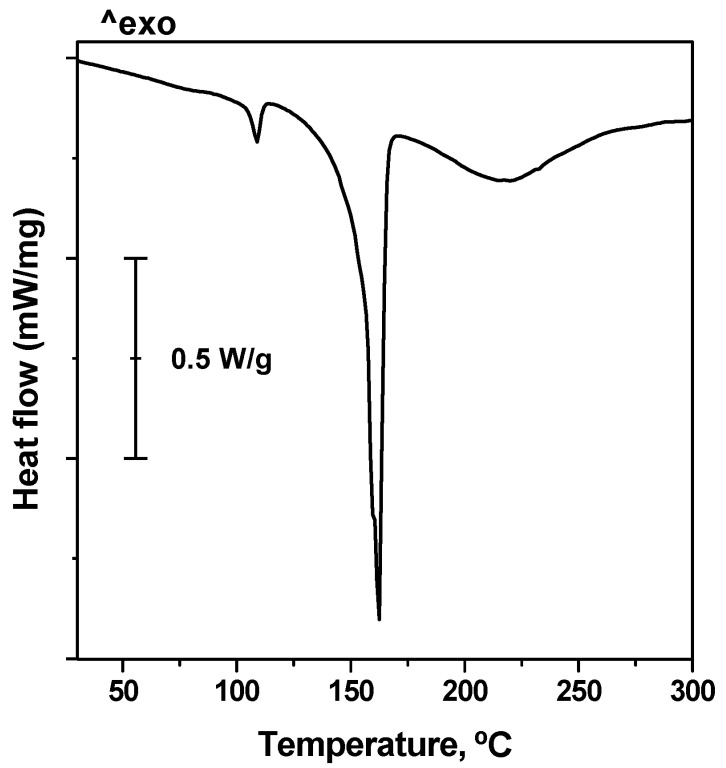
DSC profile of the solid residue of the glycolysis reaction.

**Table 1 polymers-15-04196-t001:** PET materials tested in glycolysis reactions.

PET Material	Type	Colour	Origin	Product	Content
PET R-CL	Reference	Clear	Mechanical recycling	Bottles	Water
PET C-BL	Highly coloured	Blue	Post-consumer waste	Bottles	Water
PET C-GY	Highly coloured	Grey	Post-consumer waste	Bottles	Soft drink
PET C-GN	Highly coloured	Green	Post-consumer waste	Bottles	Soft drink
PET C-WH	Highly coloured	White	Post-consumer waste	Bottles	Various
PET C-AM	Highly coloured	Amber	Post-consumer waste	Packages	Miscellaneous
PET M-AM	Multi-layered	Amber	Post-consumer waste	Bottles	Beer

**Table 2 polymers-15-04196-t002:** Glycolysis reactions with reference PET and post-consumer waste PET.

Run	t (h)	T (°C)	PET Material	PET (g)	EG/PET (mol/mol)	Zn(OAc)_2_/PET (*w*/*w*)	BHET Yield (%)
#1	8	196	PET R-CL	30	7.6/1	1%	67.1
#2	8	196	PET R-CL	30	7.6/1	---	56.8
#3	1	165	PET R-CL	30	7.6/1	1%	39.4
#4	1	180	PET R-CL	30	7.6/1	1%	66.1
#5	1	196	PET R-CL	30	3.8/1	1%	54.9
#6	1	196	PET R-CL	30	5.7/1	1%	66.1
#1	2	196	PET R-CL	30	7.6/1	1%	67.1
#7	2	196	PET C-BL	30	7.6/1	1%	63.9
#8	2	196	PET C-GY	30	7.6/1	1%	67.3
#9	2	196	PET C-GN	30	7.6/1	1%	65.8
#10	2	196	PET C-WH	30	7.6/1	1%	61.2
#11	2	196	PET C-AM	30	7.6/1	1%	67.5
#12	2	196	PET M-AM	30	7.6/1	1%	57.2

## Data Availability

The data presented in this study are available on request from the corresponding author.

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
