# Peer review of "Viability of Glycolysis for the Chemical Recycling of Highly Coloured and Multi-Layered Actual PET Wastes"

_polymers, 2023, doi:10.3390/polym15204196_

Round 1

Reviewer 1 Report

The manuscript entitled “Solvolysis of on-line identified and refused highly coloured and complex PET post-consumer waste from bottles and packages” has been reviewed. The manuscript was not well prepared. Detailed comments are as follows:

1.      The abbreviations, such as PET, rPET, FTIR, DSC, NMR and HDPE, should be given where their full names they first mentioned in the main text.

2.      Avoid using the full capitalization of first letters in the full names of NIR and MIR.

3.      There are some typo errors. For e.g., Infra-Red in Line 130 should be infrared. 1wt% in Line 223 should be 1 wt%. Please double check the manuscript.

4.      The name of purge gas flow for DSC measurement should be provided.

5.      For NIR spectra in Figs. 1 and 2, Intensity should be Absorbance.

6.      In Fig. 7, the name and unit of y-axis should be Heat flow (a.u.).

7.      In Fig. 9, the name and unit of y-axis should be Heat flow (mW). In addition, the ticks of y-axis should be included.

8.      Literatures in the main text should be marked with numbers, not surname + year.

9.      References should be revised as per guide for authors of Polymers and be checked items by items. Pay more attention to the following errors:

1)        Ref. 1 was repeated.

2)        Doi should be removed. (e.g. Ref. 6).

3)        The abbreviation of journal names should be unified. (e.g. Ref. 11).

4)        Pay attention to the capitalization of all first letters in article names. (e.g. Ref. 10).

5)        The position of year in the references should be unified. (e.g. Refs. 14 and 15).

6)        Ref. 24 and Ref. 25 is the same reference.

The manuscript entitled “Solvolysis of on-line identified and refused highly coloured and complex PET post-consumer waste from bottles and packages” has been reviewed. The manuscript was not well prepared. Detailed comments are as follows:

1.      The abbreviations, such as PET, rPET, FTIR, DSC, NMR and HDPE, should be given where their full names they first mentioned in the main text.

2.      Avoid using the full capitalization of first letters in the full names of NIR and MIR.

3.      There are some typo errors. For e.g., Infra-Red in Line 130 should be infrared. 1wt% in Line 223 should be 1 wt%. Please double check the manuscript.

4.      The name of purge gas flow for DSC measurement should be provided.

5.      For NIR spectra in Figs. 1 and 2, Intensity should be Absorbance.

6.      In Fig. 7, the name and unit of y-axis should be Heat flow (a.u.).

7.      In Fig. 9, the name and unit of y-axis should be Heat flow (mW). In addition, the ticks of y-axis should be included.

8.      Literatures in the main text should be marked with numbers, not surname + year.

9.      References should be revised as per guide for authors of Polymers and be checked items by items. Pay more attention to the following errors:

1)        Ref. 1 was repeated.

2)        Doi should be removed. (e.g. Ref. 6).

3)        The abbreviation of journal names should be unified. (e.g. Ref. 11).

4)        Pay attention to the capitalization of all first letters in article names. (e.g. Ref. 10).

5)        The position of year in the references should be unified. (e.g. Refs. 14 and 15).

6)        Ref. 24 and Ref. 25 is the same reference.

Author Response

Thanks to the reviewer's contributions, the manuscript has been substantially improved according to the indications provided. Please see the attachment.

Reviewer 2 Report

It is a useful manuscript. I suggest the following minor corrections.

The title is clumsy and inappropriate, it should be abridged and changed

The Introduction section is too long. It should be a formulation of a specific task and not a review of the literature. 

I'd like to see in the revised manuscript a chemical reaction of glycolysis. What is the mechanism of zinc acetate action? Let it be even the authors' speculation.

Zn(OAc)2 not Zn(Ac)2.

Why do the authors stress that they dealt with highly colored materials? Even in the present title! 

Author Response

(The authors gave the same response as above.)

Reviewer 3 Report

Overview

In the present manuscript the authors reported an efficient strategy to recover bis(hydroxyethyl) terephthalate (BHET) from polyethylene terephthalate (PET) post-consumer waste. In particular, PET was separated from other packaging waste by using an automatic sorting system based on Near Infrared (NIR) spectroscopy identification and depolymerized by glycolysis in ethylene glycol and zinc acetate as a catalyst. After purification, BHET was finally obtained in high yield and purity, independently of the type or colour of PET waste. Thermal and spectroscopic analyses nicely corroborate the reported findings. Overall, the work largely confirms previous reports on the chemical recycling of PET by using the same approach.

General comments

In the context of chemical recycling of PET, the novelty of the present study is not clearly depicted. A largely similar process in terms of solvent, catalyst, type of PET waste (coloured and multilayer) and its optimization, was already described in two previous reports from the same authors, coming to closely similar findings in terms of BHET yield and purity:

1)      R. López-Fonseca, I. Duque-Ingunza, B. de Rivas, S. Arnaiz, J.I. Gutiérrez-Ortiz, Polymer Degradation and Stability, Volume 95, Issue 6, 2010, Pages 1022-1028

2)      A. Aguado, L. Martínez, L. Becerra, M. Arieta-araunabeña, S. Arnaiz, A. Asueta, et al., Journal of Material Cycles and Waste Management 2014 Vol. 16 Issue 2 Pages 201-210

Differently from the above cited studies, a commercially available automatic sorting equipment based on NIR is employed, which is already established technology in plastic waste separation (see for example), and provides a similar PET quality waste, i.e., coloured and multilayer. Therefore, authors should acknowledge in the manuscript those studies and better evidence the progress beyond the state of art in order to improve the overall significance of the study.

Specific comments

Line 187. The authors indicate that PET spectrum has a distinct peak at 1657 nm in the NIR spectra reported in Figure 1 and Figure 2. For better clarity, the peak should be highlighted in the graphs.

Line 190-191 Figure 1 and 2. The authors should specify if the intensity on the y axis is absorbance, transmittance or reflectance in order to clearly read the spectra.

Line 244. 1H-NMR spectrum of recovered BHET is not reported in the manuscript. Authors should include it in the manuscript for better clarity

Line 260. Figure 8. For clarity, the frequency scale on the x axis should be reversed, so that wavenumber decreases from left (around 4000 cm-1) to right (500 cm-1).

Lines 363-366 Double reference

Author Response

(The authors gave the same response as above.)

Round 2

Reviewer 1 Report

The manuscript has been well revised. It can be accepted if the following comments are considered:

1. In all DSC thermograms, the direction of exothermic or endothermic direction should be marked.

2. In Fig. 10, only one DSC thermogram is shown. Therefore, the ticks and numbers on y-axis should be included and the unit (a.u.) should be mW or mW/mg.

3. References should be checked again. Pay attention to the following errors:

(1) the abbreviations of journal names, e.g., Ref. 9.

(2) published place and publisher, e.g., Ref. 15.

Reviewer 3 Report

The manuscript has been thoroughly revised. Minor issue should be solved:

Figure 1 and Figure 2. The authors reported a distinct peak at 1657 nm in the NIR spectra. Since the peak is pointing downwards, the label on the y axis is not absorbance, but reflectance. See for example the PET spectrum reported in (Sensors 201515(1), 2205-2227 )

Please correct the label accordingly to improve clarity.
